# Changes in daily mental health service use and mortality at the commencement and lifting of COVID-19 'lockdown' policy in 10 UK sites: a regression discontinuity in time design

Ioannis Bakolis,[1] Robert Stewart [ID] ,[2,3] David Baldwin,[4,5] Jane Beenstock,[6] Paul Bibby,[6] Matthew Broadbent,[2] Rudolf Cardinal [ID] ,[7,8] Shanquan Chen,[7] Karthik Chinnasamy,[9] Andrea Cipriani [ID] ,[10,11] Simon Douglas,[12] Philip Horner,[6] Caroline A Jackson [ID] ,[13] Ann John [ID] ,[14] Dan W Joyce,[15] Sze Chim Lee,[14] Jonathan Lewis,[16] Andrew McIntosh [ID] ,[17] Neil Nixon,[18,19] David Osborn,[20] Peter Phiri,[4] Shanaya Rathod,[4] Tanya Smith,[11] Rachel Sokal,[19] Rob Waller,[21] Sabine Landau[1]

IB and RS contributed equally.

For numbered affiliations see end of article.

**Correspondence to**
Dr Robert Stewart;
r.stewart@iop.kcl.ac.uk

## ABSTRACT

**Objectives** To investigate changes in daily mental health (MH) service use and mortality in response to the introduction and the lifting of the COVID-19 'lockdown' policy in Spring 2020.

**Design** A regression discontinuity in time (RDiT) analysis of daily service-level activity.

**Setting and participants** Mental healthcare data were extracted from 10 UK providers.

**Outcome measures** Daily (weekly for one site) deaths from all causes, referrals and discharges, inpatient care (admissions, discharges, caseloads) and community services (face-to-face (f2f)/non-f2f contacts, caseloads): Adult, older adult and child/adolescent mental health; early intervention in psychosis; home treatment teams and liaison/Accident and Emergency (A&E). Data were extracted from 1 Jan 2019 to 31 May 2020 for all sites, supplemented to 31 July 2020 for four sites. Changes around the commencement and lifting of COVID-19 'lockdown' policy (23 March and 10 May, respectively) were estimated using a RDiT design with a difference-in-difference approach generating incidence rate ratios (IRRs), meta-analysed across sites.

**Results** Pooled estimates for the lockdown transition showed increased daily deaths (IRR 2.31, 95% CI 1.86 to 2.87), reduced referrals (IRR 0.62, 95% CI 0.55 to 0.70) and reduced inpatient admissions (IRR 0.75, 95% CI 0.67 to 0.83) and caseloads (IRR 0.85, 95% CI 0.79 to 0.91) compared with the pre lockdown period. All community services saw shifts from f2f to non-f2f contacts, but varied in caseload changes. Lift of lockdown was associated with reduced deaths (IRR 0.42, 95% CI 0.27 to 0.66), increased referrals (IRR 1.36, 95% CI 1.15 to 1.60) and increased inpatient admissions (IRR 1.21, 95% CI 1.04 to 1.42) and caseloads (IRR 1.06, 95% CI 1.00 to 1.12) compared with the lockdown period. Site-wide activity, inpatient care and community services did not return to pre lockdown levels after lift of lockdown, while number of deaths did.

### Strengths and limitations of this study

► The data were obtained from a large number of mental health providers covering an extensive and varied geography within the UK: we believe, this is the most extensive multisite evaluation to date.

► Despite this, data are catchment specific and would require further replication to clarify national and international generalisability.

► Data are combined across a number of potentially diverse services for each site, with no attempt to investigate within-site heterogeneity.

► The changes described here evaluated the first wave of the COVID-19 'lockdown' policy in the UK and cannot necessarily be extrapolated to further lockdown periods.

Between-site heterogeneity most often indicated variation in size rather than direction of effect.

**Conclusions** MH service delivery underwent sizeable changes during the first national lockdown, with as-yet unknown and unevaluated consequences.

## INTRODUCTION

The 'first wave' of the COVID-19 pandemic affected healthcare sectors, not only through the virus' direct effects on communities and healthcare staff but also from the national public health policies enacted to reduce spread.[1] Mental health (MH) care faced a range of challenges in many countries, including the heightened vulnerability of its patient populations (eg, through cardiovascular and respiratory disorders), already-reduced life expectancy[2] and problems in

accessing healthcare.[3 4] Services had to be radically reconfigured to manage suspected or confirmed COVID-19 infections in inpatient and outpatient settings, staff sickness or self-isolation, the need to minimise face-to-face (f2f) contacts and the imperative to accommodate increasing pressures on acute medical care from cases of viral infection.[5 6] In turn, these changes were accompanied by the (as yet unknown) impacts of 'social distancing' on already isolated or otherwise vulnerable populations (although with potential positive effects of increased support, sense of community cohesion and reduced social strain for some individuals) and of economic constraints on already impoverished and disadvantaged communities.[7 8] This indicates a pressing need for research assessing the impact of the pandemic and lockdown policies on MH service demand and uptake.[9]

Published data are emerging on MH outcomes in community samples[10] and specific groups such as healthcare workers,[11] as well as on infection rates in MH inpatients,[12] and staff/expert concerns about MH provision.[6 13] However, there have been few studies to date quantifying changes in mental healthcare activity.[5] Single-site reports from UK services have highlighted falls in activity following the 23 March[d] national lockdown, followed by increased demand for some services, decreased activity for others, shifts from f2f to virtual consultations and a rise in mortality.[14–19] Drawing on a network of providers, we sought to determine the level and heterogeneity of such mental healthcare changes across multiple national sites using a quasi-experimental, regression discontinuity design in time.

## METHODS
### Data sources
Early in the first UK lockdown, we enquired of several MH care providers as to the feasibility of a relatively short-notice extraction of service-level data according to a standardised protocol. Sites were sourced initially from the National Institute for Health Research (NIHR) Mental Health Translational Research Collaboration, a network of MH services with NIHR or equivalent funding. MH trusts in England provide all National Health Service (NHS) specialist MH care to defined geographic catchment areas. Information from these was supplemented by further extractions of available mental healthcare data from Scotland and Wales service providers.

The following English trusts were able to participate: (1) Cambridgeshire and Peterborough Foundation Trust, serving the Combined Authority of Cambridgeshire and Peterborough, around 860 000 people; (2) Camden & Islington Foundation Trust, serving the Camden and Islington boroughs of north London, around 500 000 people; (3) Cumbria, Northumberland, Tyne & Wear Foundation Trust, based in Newcastle and covering more than 70 sites across North Cumbria, Northumberland and Tyne and Wear, a population of around 2 million; (4) Lancashire & South Cumbria Foundation Trust,

serving Lancashire and South Cumbria, population 1.8 million; (5) Oxford Health Foundation Trust, serving Oxfordshire, Buckinghamshire, Wiltshire, Milton Keynes, Swindon, Bath and North East Somerset, a population of around 3 million; (6) Nottinghamshire Healthcare Trust, serving City and County of Nottinghamshire, around 1.2 million people; (7) South London and Maudsley Foundation Trust, serving Croydon, Lambeth, Lewisham and Southwark boroughs of south London, a population of around 1.3 million and (8) Southern Health Foundation Trust, serving Hampshire (excluding Portsmouth City), a population of around 1.3 million. Additional data were made available from NHS Lothian, a provider of all health services to a population of 850 000 in and around Edinburgh, including all physical and MH services and from the Secure Anonymised Information Linkage databank which contains anonymised data on all health services provided to the population of Wales, around 3 million.

All data were extracted at service level, with no individual patient-level extractions. Sites were deidentified prior to analysis and are referred to subsequently as sites A–J.

Overall, 9 of the 10 sites extracted daily activity data from 1 January 2019 to 31 May 2020; extractions took place in June and July 2020. Overall, 4 of the 10 sites were subsequently able to provide extensions of daily data up to 31 July 2020, extracted in August and September 2020. One site extracted (in September 2020) all activity data on a weekly level from 1 January 2020 to 5 July 2020 in addition to 2019.

### MH service activity and mortality variables
Where possible, sites extracted the following data for each day:
1. Site activity: number of new referrals accepted and number of discharges from services.
2. MH inpatient services: number of new admissions; number of discharges and daily inpatient caseload (total and caseload detained under the Mental Health Act (MHA) section for all age groups).
3. Community services: For each service, the number of f2f contacts, non-f2f contacts, total contacts (sum of f2f and non-f2f), cancelled appointments or non-attendance (DNA) and daily caseload (patients with an active referral to that service). These were extracted for each of the following community service types:
   a. Adult MH (AMH; community MH teams).
   b. Child and adolescent MH services (CAMHS).
   c. Early intervention in psychosis (EIP).
   d. Home treatment team (HTT)—also known as crisis resolution teams which provide the option for at least daily reviews as an alternative to inpatient care.
   e. Liaison psychiatry services, including services provided to emergency departments and settings created as alternatives to emergency departments during the pandemic.
   f. Older adults (OA) MH services.

Daily mortality was extracted as recorded within clinical systems from routine data supplied to sites from national sources. For sites A, B, E, F, H and J, deaths recorded on the clinical system represent those for all past service contacts regardless of whether the individual is a current patient at the time of death (for 90% of the catchment served by site H). Sites D and G ascertain deaths for all current patients at the time of death along with selected discharged patients (up to 6 months post discharge for site G). For site C, deaths were restricted to those on MH inpatient wards at the time of death. For site I, weekly deaths and weekly inpatient admissions/discharges were extracted where MH was recorded as the primary diagnosis.

## Statistical analysis

To describe changes in the above measures, we used two time points coinciding with the commencement and lifting of the COVID-19 'lockdown' policy: 23 March 2020 and 10 May 2020, respectively. To estimate these, we deployed a regression discontinuity in time (RDiT) design[20] combined with a difference-in-difference (DiD) approach for estimation, comparing measures before and after the lockdown announcement in 2020 to those before and after the same date in 2019, thus adjusting for potential seasonal changes. Similar analysis was conducted for the lift-of-lockdown announcement.

For analytical purposes, we divided the sample into two cohort periods (2019, 2020). The parameters of interest were the effects of (a) lockdown and (b) lift of the lockdown policy announcements on extracted measures in different time windows before and after 23 March 2020 and 10 May 2020, relative to that observed before and after 23 March 2019 and 10 May 2019. This approach was strengthened by borrowing elements from a DiD approach and is similar to a difference in discontinuities design because it rests on the intuition of combining an RDiT with a DiD strategy (RDiT–DiD).[21] Further rationale and details of the analytic approach are provided in the online supplemental technical appendix.

Differences in measures before versus after the cut-off dates were reported as incidence rate ratios (IRRs) and their corresponding 95% CI. Negative binomial regression models were used to assess the effect of 23 March COVID-19 'lockdown' policy on mortality and MH service activity adjusted for temporal trends (eg, weekday, month, year) and taking into account overdispersion (please technical appendix and online supplemental figures 1–5). Results from each site extracting daily data were pooled across sites using random effects meta-analysis and heterogeneity was summarised using the $I^2$ statistic.[22]

A series of sensitivity analyses were also performed.

1. We reran a subgroup analysis of our main RDiT–DiD analysis for four sites with daily data that could update their data extractions to 31 July 2020. We increased the time window after the lift of the lockdown announcement in order to capture any potential effects of the lift of the lockdown announcement on mortality and MH service activity with greater statistical certainty due to larger numbers.

2. To account for anticipatory effects of the lockdown and lift-of-lockdown announcements, we omitted 1 week either side of the cut-off date of 23 March 2020 (ie, 16 March 2020 to 30 March 2020) and 1 week either side of 10 May 2020 (ie, 3 May 2020 to 17 May 2020). Anticipatory effects, such as the announcement of social distancing on 16 March 2020, could have resulted in altered MH service activity in the study population just before the implementation of lockdown. Similarly, non-adherence to the lockdown rules was documented during the month of April 2020 and before lockdown was lifted.[23]

3. To account for changes in the infection rate, we further adjusted our models for national COVID-19-related deaths occurring 1 week prior to the date of our observed daily counts.

4. To check robustness of our findings, that these were not an artefact of the pre lockdown time period from 1 January 2020, we reduced the time window before the lockdown announcement from 4 February 2020. Thus, we used the same number of daily observations before and after the lockdown announcement (23 March to 10 May 2020 vs 4 February to 23 March 2020) and the lift-of-lockdown announcement (10 May to 31 May 2020 vs 19 February to 23 March 2020).

5. To account for changes due to the media coverage of the pandemic, we further adjusted our models for trends in the UK occurring on the day of our observed daily counts. Google Trends data provide an unfiltered sample of search requests made to Google. It supplies an index for search intensity by topic over the time period requested in the UK area. This is the number of daily searches for the specified topic divided by the maximum number of daily searches for this topic over the time period in question in the UK. This is scaled from 0 to 100. A value of 100 is the peak popularity for the term. A value of 50 means that the term is half as popular. A score of 0 means there were not enough data for this term.[24]

Analyses were performed using STATA V.15.1 (Stata Corporation, College Station, Texas, USA).

## RESULTS

The 10 participating sites are described and compared in table 1 and pooled IRRs from the random-effect meta-analysis are presented in table 2 for the three period comparisons of interest in the sites extracting daily data.

### Transition to lockdown

For this transition, pooled estimates showed a significant and greater than twofold increase in daily deaths (IRR 2.31 (95% CI: 1.86 to 2.87); figure 1). New accepted referrals decreased (IRR 0.62 (95% CI: 0.55 to 0.70)). Pooled estimates from inpatient care showed a decrease in new admissions (IRR 0.75 (95% CI: 0.67 to 0.83)), total daily

**Table 1** Descriptive characteristics of daily mortality and mental health (MH) service caseloads by site

| Measures | Site | Before lockdown* Median (IQR) | During lockdown* Median (IQR) | After lockdown* Median (IQR) |
|---|---|---|---|---|
| Number of deaths | A | 13 (11–17) | 30 (22–38) | 13 (11–16) |
| | B | 44 (39–51) | 67 (61–78) | 40 (29–47) |
| | C | 0 (0–0) | 0 (0–1) | 0 (0–0) |
| | D | 25 (22–28) | 42 (28–50) | 7 (4–18) |
| | E | 2 (0–3) | 3 (0–6) | 0 (0–1) |
| | F | 12 (9–14) | 18 (12–21) | 13 (10–16) |
| | G | 7 (6–9) | 10 (7–12) | 5 (4–7) |
| | H | 3 (2–4) | 4 (3–6) | 3 (2–3) |
| | I | 95 (93–100)† | 130 (112-146)† | 85 (80–88)† |
| | J | 10 (7–13) | 18 (14–26) | 11 (8–12) |
| Inpatient caseload | A | 768 (762–774) | 556 (544–567) | 581 (568–588) |
| | B | 454.5 (445–461) | 363 (357–368) | 375 (374–379) |
| | C | 495 (489–503) | 427.5 (422–443.5) | 451 (443–454) |
| | D | 701 (696–707) | 625 (617–657) | 623 (613–640) |
| | E | 481 (471–491) | 412 (400–418) | 453 (437–463) |
| | F | 207 (205–211) | 140 (135–146) | 147 (146–148) |
| | H | 408 (399–412) | 351 (338–360) | 367 (358–381) |
| | J | 400 (394–407) | 361 (354–371) | 375 (365–385) |
| AMH (community) caseload | A | 8691 (8666–8714) | 8504 (8438–8576) | 8398 (8389–8409) |
| | B | 8700 (8652–8739) | 8039 (7823–8297) | 7555 (7498–7587) |
| | D | 11 039.5 (10 956–11091) | 10 803 (10 688–10 923) | 10 526 (10 520–10 559) |
| | F | 2228 (2221–2242) | 2178 (2167–2195) | 2158 (2156–2162) |
| | H | 724.5 (719–726) | 695 (686–700) | 684 (683–686) |
| | J | 10 687 (10 506–10 981) | 10 160 (9866–10 519) | 9599 (9529–9612) |
| CAMHS caseload | A | 6915.5 (6792–7021) | 6962 (6875–6997) | 6789 (6752–6801) |
| | D | 10 240.5 (10 187–10 292) | 10 060 (9935–10 189) | 9807 (9795–9813) |
| | E | 38 (37–38) | 35 (35–35) | 35 (35–35) |
| | F | 2232 (2176–2247) | 2106 (2061–2144) | 1992 (1988–1999) |
| | H | 4375.5 (4347–4445) | 4383 (4234–4414) | 4106 (4078–4137) |
| | J | 15 946.5 (15 610–16 046) | 14 821 (14 342–15 394) | 13 835 (13 802–13 942) |
| EIP caseload | A | 1193 (1176–1208) | 1190 (1185–1198) | 1206 (1199–1208) |
| | B | 354.5 (347–358) | 333 (329–343) | 336 (335–341) |
| | D | 809 (805–812) | 794 (790–806) | 806 (804–810) |
| | E | 862 (857–867) | 852 (849–853) | 860 (859–862) |
| | F | 93 (92–96) | 97 (96–98) | 95 (93–95) |
| | H | 768 (765–771) | 767 (762–771) | 765 (763–768) |
| | J | 421 (415–424) | 421 (419–423) | 421 (415–424) |
| HTT caseload | A | 214.5 (207–222) | 153 (148–158) | 191 (187–197) |
| | B | 258.5 (241–276) | 151 (139–161) | 198 (188–202) |
| | D | 263 (247–289) | 162 (151–177) | 206 (197–214) |
| | E | 219 (202–234) | 95 (87–103) | 127 (121–133) |
| | F | 73 (64–77) | 54 (50–62) | 56 (50–59) |
| | H | 977.5 (968–996) | 780 (715–850) | 732 (725–740) |
| | J | 354.5 (293–418) | 100 (60–165) | 50 (48–51) |

Continued

**Table 1** Continued

| Measures | Site | Before lockdown* Median (IQR) | During lockdown* Median (IQR) | After lockdown* Median (IQR) |
|---|---|---|---|---|
| Liaison caseload | A | 1060 (1044–1074) | 971 (956–980) | 1005 (999–1015) |
| | B | 80.5 (69–91) | 27 (21–29) | 47 (39–51) |
| | D | 1851.5 (1786–1879) | 1523 (1494–1573) | 1505 (1493–1522) |
| | E | 103 (90–110) | 48 (42–65) | 81 (75–86) |
| | F | 499 (468–516) | 407 (403–411) | 448 (436–455) |
| | H | 59.5 (57–62) | 64 (63–68) | 71 (68–74) |
| | J | 221 (216–227) | 203 (193–211) | 200 (197–202) |
| OA caseload | A | 1297 (1264–1308) | 1065 (1018–1126) | 1001 (991–1007) |
| | B | 6243 (6193–6257) | 5518 (5336–5746) | 5149 (5132–5191) |
| | D | 5731.5 (5677–5791) | 5497 (5398–5596) | 5346 (5338–5352) |
| | E | 567 (561–582) | 503 (497–508) | 517 (511–518) |
| | F | 502 (487–507) | 350 (340–386) | 344 (342–345) |
| | H | 1638 (1621–1654) | 1328 (1288–1383) | 1261 (1259–1263) |
| | J | 3844 (3821–3857) | 3426 (3337–3557) | 3284 (3280–3286) |

*Before lockdown: 1 January 2020 to 22 March 2020; during lockdown: 23 March 2020 to 9 June 2020; after lockdown: 10 May 2020 to 31 May 2020.
†For site I, we ascertained median and IQR for weekly mortality and have divided each estimate here by 7 for comparability.
AMH, adult MH service; CAMHS, child and adolescent MH service; EIP, early intervention for psychosis service; HTT, home treatment team; OA, older adult service.

caseload (IRR 0.85 (95% CI: 0.79 to 0.91)) and numbers of patients detained under an MHA section (IRR 0.93 (95% CI: 0.87 to 0.99)). Of community services evaluated, all showed sizeable decreases in daily numbers of f2f contacts and increases in non-f2f contacts (both strongest for EIP). Total contacts reduced in liaison and HTT services and increased (but not significantly so) in AMH, CAMHS, EIP and OA services. Daily caseloads significantly reduced in CAMHS, HTT and liaison and significantly increased in AMH. No significant changes were observed in the numbers of cancelled or non-attended appointments, apart from a reduction within HTT.

### Transition from lockdown

Numbers of deaths reduced considerably (IRR 0.42 (95% CI: 0.27 to 0.66)), and IRR estimates for post lockdown versus pre lockdown differences were not significantly different (IRR 0.97 (95% CI: 0.68 to 1.38)). Numbers of new site referrals increased after the lifting of lockdown (IRR 1.36 (95% CI: 1.15 to 1.60)) compared with the lockdown period but remained lower than pre lockdown (IRR 0.85 (95% CI: 0.76 to 0.95)). Inpatient services saw increased admissions (IRR 1.21 (95% CI: 1.04 to 1.42)) and reduced discharges (IRR 0.66 (95% CI: 0.53 to 0.84)) after the lifting of lockdown. Discharges remained lower than before lockdown, as did caseloads. F2f contacts increased after lockdown in all services apart from AMH and CAMHS, increases in non-f2f contacts were seen in AMH, EIP and OA services and total contacts increased in EIP, HTT, liaison and OA services. Compared with pre lockdown, pooled daily f2f contacts reduced and non-f2f

contacts increased in all services after lifting of lockdown apart from HTT in which f2f contacts increased and non-f2f contacts were unchanged; total contacts increased in AMH, CAMHS and EIP services and decreased in liaison. Daily caseloads decreased in HTT and liaison services and cancelled or non-attended appointments reduced in liaison services.

There was moderate-to-substantial heterogeneity in the majority of our IRRs estimates at the three transitions in number of deaths, inpatient caseloads, f2f and non-f2f contacts and daily caseloads for all community services. For example, IRRs for number of deaths on transition into lockdown ranged from 1.55 (95% CI: 1.23 to 1.94) to 5.30 (95% CI: 2.87 to 9.79) (figure 1). All IRRs are provided individually by site in online supplemental tables 1–3. Inpatient caseload reductions on the transition into lockdown varied in IRRs from 0.67 (95% CI: 0.65 to 0.69) to 1.00 (95% CI: 0.99 to 1.01), reductions in HTT caseloads from 0.42 (95% CI: 0.39 to 0.45) to 0.98 (95% CI: 0.97 to 0.99) and liaison caseload reductions from 0.35 (95% CI: 0.31 to 0.39) to 1.03 (95% CI: 1.00 to 1.06). IRRs for the transition from before lockdown to lifting of lockdown (online supplemental table 3) also varied, with daily deaths ranging from 0.31 (95% CI: 0.22 to 0.46) to 1.64 (95% CI: 1.29 to 2.09), those for inpatient caseloads from 0.87 (95% CI: 0.85 to 0.89) to 1.08 (95% CI: 1.07 to 1.09), HTT caseloads from 0.17 (95% CI: 0.16 to 0.18) to 0.91 (95% CI: 0.89 to 0.92) and liaison caseloads from 0.41 (95% CI: 0.36 to 0.47) to 1.06 (95% CI: 1.03 to 1.10).

**Table 2** Estimated effects of lockdown and lift-of-lockdown announcements on service measures meta-analysed across sites A–H and J from 1 January to 31 May 2020, estimating transitions related to the lockdown announcement (23 March) and lift-of-lockdown announcement (10 May)

| Measures | Lockdown announcement versus pre lockdown IRR (95% CI) | I² | Lift of lockdown announcement versus lockdown IRR (95% CI) | I² | Lift of lockdown versus before lockdown IRR (95% CI) | I² |
|---|---|---|---|---|---|---|
| Number of deaths | 2.31** (1.86 to 2.87) | 85.3 | 0.42** (0.27 to 0.66) | 91.8 | 0.97 (0.68 to 1.38) | 88.5 |
| **Trust-wide activity** | | | | | | |
| Number of new referrals accepted | 0.62** (0.55 to 0.70) | 0.0 | 1.36** (1.15 to 1.60) | 0 | 0.85** (0.76 to 0.95) | 0.0 |
| Number of discharges | 0.95 (0.85 to 1.06) | 0.0 | 0.95 (0.81 to 1.11) | 0 | 0.89 (0.80 to 1.00) | 0.0 |
| **Inpatient care** | | | | | | |
| New admissions | 0.75** (0.67 to 0.83) | 0.0 | 1.21** (1.04 to 1.42) | 0 | 0.90 (0.79 to 1.03) | 0.0 |
| Discharges | 1.13 (0.97 to 1.31) | 0.0 | 0.66** (0.53 to 0.84) | 5.2 | 0.75** (0.62 to 0.90) | 0.0 |
| Inpatient caseload | 0.85** (0.79 to 0.91) | 99.6 | 1.06 (1.00 to 1.12) | 99.0 | 0.90** (0.83 to 0.97) | 99.7 |
| Inpatient caseload on a MHA section | 0.93* (0.87 to 0.99) | 99.4 | 1.01 (0.93 to 1.10) | 99.6 | 0.94 (0.86 to 1.02) | 99.8 |
| **AMH (community)** | | | | | | |
| F2f contacts | 0.55** (0.44 to 0.68) | 49.4 | 0.99 (0.80 to 1.23) | 0 | 0.57** (0.44 to 0.75) | 40.2 |
| Non-f2f contacts | 3.74** (2.58 to 5.43) | 83.2 | 1.27** (1.06 to 1.52) | 0 | 3.95** (2.73 to 5.71) | 70.3 |
| F2f and non-f2f contacts | 1.07 (0.94 to 1.22) | 0.0 | 1.12 (0.94 to 1.34) | 0 | 1.19** (1.04 to 1.36) | 0.0 |
| Cancelled appointments or DNAs | 0.85 (0.61 to 1.18) | 73.1 | 1.02 (0.80 to 1.30) | 10.6 | 0.79 (0.56 to 1.11) | 61.7 |
| Caseload | 1.03* (1.01 to 1.05) | 98.2 | 1.01 (1.00 to 1.02) | 94.6 | 1.04** (1.02 to 1.04) | 0.0 |
| **CAMHS** | | | | | | |
| F2f contacts | 0.35** (0.17 to 0.74) | 93.2 | 1.12 (0.79 to 1.61) | 14.5 | 0.36** (0.15 to 0.88) | 88.8 |
| Non-f2f contacts | 3.58** (2.54 to 5.04) | 70.2 | 1.23 (1.00 to 1.50) | 0 | 3.76** (2.69 to 5.25) | 50.1 |
| F2f and non-f2f contacts | 1.08 (0.93 to 1.26) | 0.0 | 1.20 (0.96 to 1.49) | 0 | 1.23** (1.04 to 1.47) | 0.0 |
| Cancelled appointments or DNAs | 0.87 (0.67 to 1.14) | 48.8 | 0.94 (0.65 to 1.36) | 41.2 | 0.89 (0.76 to 1.04) | 0.0 |
| Caseload | 0.93* (0.89 to 0.97) | 99.2 | 1.03 (0.98 to 1.08) | 99.3 | 0.94 (0.88 to 1.01) | 99.5 |
| **EIP** | | | | | | |
| F2f contacts | 0.27** (0.18 to 0.42) | 86.0 | 1.64** (1.19 to 2.27) | 48.4 | 0.45** (0.29 to 0.70) | 82.5 |
| Non-f2f contacts | 4.04** (2.63 to 6.20) | 83.3 | 1.48** (1.03 to 2.12) | 52.9 | 5.51** (3.08 to 9.84) | 87.5 |
| F2f and non-f2f contacts | 1.09 (0.88 to 1.34) | 51.6 | 1.45** (1.22 to 1.72) | 0 | 1.51** (1.22 to 1.88) | 49.7 |
| Cancelled appointments or DNAs | 0.76 (0.54 to 1.08) | 74.4 | 0.99 (0.72 to 1.36) | 44.3 | 0.70 (0.44 to 1.12) | 85.2 |
| Caseload | 1.04* (1.02 to 1.06) | 97.9 | 1.04 (1.00 to 1.08) | 99.5 | 1.08** (1.03 to 1.13) | 99.8 |
| **HTT** | | | | | | |
| F2f contacts | 0.43** (0.32 to 0.57) | 97.0 | 1.49** (1.36 to 1.62) | 37.5 | 1.91** (1.49 to 2.43) | 84.3 |

Continued

**Table 2** Continued

| Measures | Lockdown announcement versus pre lockdown | | Lift of lockdown announcement versus lockdown | | Lift of lockdown versus before lockdown | |
|---|---|---|---|---|---|---|
| | IRR (95% CI) | I² | IRR (95% CI) | I² | IRR (95% CI) | I² |
| Non-f2f contacts | 1.78** (1.56 to 2.03) | 60.3 | 1.07 (0.87 to 1.32) | 69.6 | 0.99 (0.87 to 1.14) | 85.9 |
| F2f and non-f2f contacts | 0.78** (0.68 to 0.89) | 90.3 | 1.27** (1.16 to 1.39) | 52.6 | 1.08 (0.87 to 1.34) | 58.3 |
| Cancelled appointments or DNAs | 0.68** (0.51 to 0.89) | 85.6 | 1.47** (1.11 to 1.96) | 67.9 | 0.62** (0.43 to 0.88) | 99.8 |
| Caseload | 0.62** (0.49 to 0.78) | 99.7 | 1.01 (0.78 to 1.29) | 99.4 | 0.64** (0.51 to 0.82) | 94.4 |
| Liaison | | | | | | |
| F2f contacts | 0.43** (0.36 to 0.50) | 78.2 | 1.67** (1.39 to 2.00) | 59.0 | 0.72** (0.61 to 0.86) | 66.3 |
| Non-f2f contacts | 1.50** (1.15 to 1.96) | 68.8 | 1.11 (0.81 to 1.53) | 52.5 | 1.69** (1.16 to 2.47) | 74.2 |
| F2f and non-f2f contacts | 0.60** (0.54 to 0.66) | 44.5 | 1.46** (1.29 to 1.65) | 28.3 | 0.85* (0.74 to 0.98) | 56.7 |
| Cancelled appointments or DNAs | 0.62 (0.35 to 1.10) | 84.5 | 0.75 (0.46 to 1.24) | 45.9 | 0.65** (0.43 to 0.97) | 41.6 |
| Caseload | 0.75** (0.69 to 0.81) | 99.1 | 1.14** (1.07 to 1.21) | 97.7 | 0.88** (0.83 to 0.93) | 97.9 |
| OA | | | | | | |
| F2f contacts | 0.32** (0.20 to 0.49) | 83.8 | 1.45** (1.13 to 1.85) | 0.35 | 0.39** (0.23 to 0.67) | 82.3 |
| Non-f2f contacts | 3.66** (2.54 to 5.27) | 78.7 | 1.28* (1.02 to 1.59) | 0 | 4.27** (2.78 to 6.56) | 73.3 |
| F2f and non-f2f contacts | 1.06 (0.81 to 1.37) | 62.5 | 1.31** (1.06 to 1.63) | 0 | 1.28 (0.94 to 1.73) | 50.8 |
| Cancelled appointments or DNAs | 1.08 (0.73 to 1.58) | 77.9 | 0.70 (0.38 to 1.31) | 75.8 | 0.80 (0.51 to 1.26) | 63.1 |
| Caseload | 0.95 (0.92 to 0.99) | 99.3 | 0.99 (0.96 to 1.01) | 98.7 | 0.93 (0.84 to 1.03) | 99.8 |
| Site I data† | | | | | | |
| Number of deaths | 1.47** (1.35 to 1.60) | | 0.72** (0.66 to 0.80) | | 1.07 (0.99 to 1.15) | |
| New inpatient admissions | 0.64** (0.58 to 0.71) | | 1.67** (1.48 to 1.88) | | 1.07 (0.93 to 1.22) | |
| Inpatient discharges | 0.83* (0.73 to 0.94) | | 1.19** (1.08 to 1.32) | | 0.99 (0.88 to 1.10) | |

Incidence rate ratios and 95% CIs and percentage heterogeneity across sites (I²) were estimated.
*p<0.05; **p<0.001.
†Derived from weekly data from 1 January to 5 July 2020.
AMH, adult mental health service; CAMHS, child and adolescent mental health service; DNA, Did not attend; EIP, early intervention for psychosis service; F2f, face to face; HTT, home treatment team; MHA, Mental Health Act; OA, older adult service.

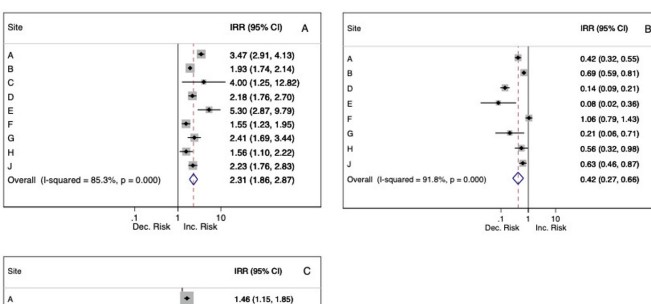

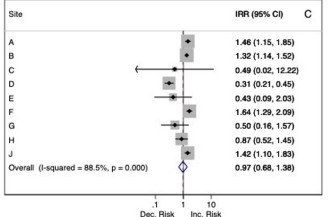

**Figure 1** Forest plots of changes in daily deaths by site associated with lockdown and lift-of-lockdown announcements, with pooled meta-analysis estimates. Incidence rate ratios (IRRs) and 95% CIs, percentage heterogeneity ($I^2$) and p values from $\chi^2$ tests of heterogeneity are displayed. (A) Initiation of lockdown announcement versus before lockdown. (B) Lift of lockdown announcement. (C) Lift of lockdown versus pre lockdown.

For site I, where extractions of numbers of deaths and service activity data were weekly rather than daily and could not therefore be combined with other sites, there was a significant increase in weekly deaths (IRR 1.47 (95% CI: 1.35 to 1.60)) at lockdown and a decrease (IRR 0.72 (95% CI: 0.66 to 0.80)) after the lifting of lockdown. Slightly, but non-significantly, increased numbers of deaths were observed after the lifting of lockdown compared with before lockdown. There was a reduction in admissions (IRR 0.64 (95% CI: 0.58 to 0.71)) and discharges (IRR 0.83 (95% CI: 0.73 to 0.94)) at lockdown, but no differences after the lifting of lockdown compared with before lockdown.

### Sensitivity analyses

1. Further data for the post lockdown versus pre lockdown comparisons are presented in table 3 for the four sites able to extract daily data to 31 July. Considering the findings that were most consistently significant, all showed reduced total inpatient caseload, reduced OA f2f contacts, increased non-f2f AMH and EIP contacts and reduced CAMHS, liaison and OA caseloads.
2. When excluding data for 1 week around the lockdown and lift-of-lockdown announcements, minimal changes were observed and IRRs remained largely similar (online supplemental table 4).
3. Further adjustment for levels of national COVID-19 mortality levels (online supplemental table 5) resulted in some dilution in the strength of IRRs for deaths, new admissions and daily caseloads at lockdown initiation, but minimal changes to those IRRs that were statistically significant. A few IRRs for the lift-of-lockdown transition fell below statistical significance (increases in inpatient admissions, in OA f2f contacts), as did some

'after versus before' lockdown differences (reductions in trusts' new referrals and inpatient discharges; increases in CAMHS total contacts).
4. Additionally, no substantial changes were observed in our IRRs when we employed different equally spaced exposure time window intervals around the lockdown and lift-of-lockdown announcements (online supplemental table 6).
5. Further adjustment for Google Trends Index (online supplemental table 7) resulted in some dilution in the strength of IRRs for non-f2f contacts across services at lockdown initiation and lift-of-lockdown transition, but findings remained statistically significant. No substantial changes were observed in the other IRRs.

## DISCUSSION
### Summary of findings
To our knowledge, this is the first multisite study to present data and evaluate the impact of the COVID-19 'lockdown' policy in the UK on mortality and MH service activity provision. In summary, the initiation of lockdown was associated with increases in the death rate, decreases in new accepted referrals and decreases in inpatient admissions and daily caseload. In community services, sizeable increases were observed in non-f2f contacts, with decreases in f2f contacts. Caseloads decreased in CAMHS, HTT, liaison and OA services, but increased slightly in AMH and EIP services. The lift-of-lockdown transition was associated with decreased deaths, increased referrals and inpatient admissions and decreased inpatient discharges. In community settings, most services saw increases in non-f2f and total contacts. There was moderate-to-high heterogeneity of these estimates across sites; only site-wide new referrals/discharges and inpatient admissions/discharges were consistent.

### Impact of the COVID-19 first wave
The potential impact of the COVID-19 pandemic has been widely discussed in a general sense, focusing on the initial priorities of infection control, treatment options for severe complications and the preparedness of critical care services.[25] For mental healthcare, there is a need to understand the population-level impact of both the viral infection and the social distancing being imposed by many national governments,[9 26 27] as well as concerns about higher levels of population distress coupled with reduced use/availability of psychological support.[28] For people with pre-existing mental disorders, there is also a concern that vulnerability to COVID-19 infection may be higher than expected, because of infection susceptibility due to physical comorbidities and because of barriers to health service access.[29 30] A higher risk of relapse of mental disorders is a concern due to the stress of the pandemic itself, the stress of consequent quarantine,[31] reduced access to routine outpatient visits for evaluations and prescriptions and possibly also an avoidance of health services because of the perceived risk of infection or a wish to

**Table 3** Sensitivity analyses of lift-of-lockdown versus pre lockdown comparisons for the extended period of follow-up (to 31 July 2020) available for four sites

| Measures | Site A<br>Lift of the lockdown<br>versus pre lockdown<br>IRR (95% CI) | Site F<br>Lift of the lockdown<br>versus pre lockdown<br>IRR (95% CI) | Site G<br>Lift of the lockdown<br>versus pre lockdown<br>IRR (95% CI) | Site J<br>Lift of the lockdown<br>versus pre lockdown<br>IRR (95% CI) |
|---|---|---|---|---|
| Number of deaths | 1.20 (0.95 to 1.51) | 2.06** (1.09 to 3.90) | 0.89 (0.65 to 1.20) | 1.08 (0.84 to 1.40) |
| Trust-wide activity | | | | |
| Number of new referrals accepted | 0.79** (0.68 to 0.93) | 0.55* (0.30 to 0.98) | N/A | 0.73 (0.50 to 1.07) |
| Number of discharges | 0.83** (0.72 to 0.96) | 0.60 (0.31 to 1.16) | N/A | 0.74 (0.51 to 1.06) |
| Inpatient care | | | | |
| New admissions | 0.94 (0.75 to 1.18) | 0.34 (0.06 to 1.83) | N/A | 0.94 (0.62 to 1.42) |
| Discharges | 0.64** (0.48 to 0.86) | 0.24 (0.03 to 2.22) | N/A | 0.81 (0.51 to 1.28) |
| Inpatient caseload | 0.76** (0.74 to 0.77) | 0.62** (0.59 to 0.66) | N/A | 0.96* (0.95 to 0.97) |
| Inpatient caseload on a MHA section | 0.87** (0.85 to 0.88) | 1.02 (0.96 to 1.09) | N/A | 1.01 (0.99 to 1.02) |
| Adult mental health (MH) (community) | | | | |
| F2f contacts | 0.50** (0.38 to 0.67) | 0.44 (0.05 to 4.32) | 0.31** (0.18 to 0.54) | 0.47** (0.32 to 0.69) |
| Non-f2f contacts | 2.99** (2.45 to 3.64) | 12.11** (1.64 to 89.29) | 13.66** (7.77 to 24.03) | 8.69** (6.22 to 12.15) |
| F2f and non-f2f contacts | 1.28** (1.08 to 1.53) | 0.97 (0.11 to 8.55) | 0.83 (0.47 to 1.46) | 1.68** (1.16 to 2.42) |
| Cancelled appointments or DNAs | 0.70** (0.58 to 0.85) | 0.39 (0.04 to 3.53) | 0.73 (0.43 to 1.23) | 1.06 (0.67 to 1.68) |
| Caseload | 0.94* (0.91 to 0.99) | 1.04* (1.02 to 1.04) | N/A | 0.92** (0.90 to 0.93) |
| CAMHS | | | | |
| F2f contacts | 0.14** (0.08 to 0.26) | 1.70 (0.17 to 16.59) | 0.65 (0.40 to 1.03) | 0.08** (0.05 to 0.14) |
| Non-f2f contacts | 3.60** (2.96 to 4.37) | 4.78 (0.77 to 29.85) | 16.12** (9.28 to 27.99) | 3.32** (2.19 to 5.04) |
| F2f and non-f2f contacts | 1.27** (1.03 to 1.56) | 1.98 (0.22 to 17.86) | 1.15 (0.75 to 1.77) | 1.35 (0.89 to 2.05) |
| Cancelled appointments or DNAs | 0.90 (0.74 to 1.10) | 0.59 (0.06 to 5.61) | 0.46** (0.25 to 0.83) | 0.59** (0.38 to 0.93) |
| Caseload | 0.95* (0.91 to 0.97) | 0.96** (0.95 to 0.97) | N/A | 0.84** (0.82 to 0.85) |
| EIP | | | | |
| F2f contacts | 0.20** (0.16 to 0.26) | 0.79 (0.10 to 6.43) | N/A | 0.22** (0.16 to 0.31) |
| Non-f2f contacts | 4.22** (3.39 to 5.25) | 35.59** (3.38 to 74.33) | N/A | 13.90** (9.05 to 21.37) |
| F2f and non-f2f contacts | 1.39** (1.16 to 1.67) | 1.32 (0.18 to 9.73) | N/A | 1.95** (1.59 to 2.39) |
| Cancelled appointments or DNAs | 0.55** (0.43 to 0.71) | 0.86 (0.09 to 7.78) | N/A | 1.15 (0.77 to 1.73) |
| Caseload | 0.95* (0.92 to 0.98) | 1.07** (1.05 to 1.08) | N/A | 0.98* (0.96 to 0.99) |
| HTT | | | | |
| F2f contacts | 0.68** (0.61 to 0.74) | 0.99 (0.51 to 1.92) | N/A | 0.09** (0.06 to 0.13) |
| Non-f2f contacts | 2.47** (2.05 to 2.96) | 2.72** (1.04 to 7.14) | N/A | 2.32** (1.51 to 3.56) |
| F2f and non-f2f contacts | 0.99 (0.91 to 1.07) | 1.37 (0.73 to 2.56) | N/A | 0.38** (0.28 to 0.51) |
| Cancelled appointments or DNAs | 1.23 (0.97 to 1.56) | 1.33 (0.34 to 5.29) | N/A | 0.48** (0.27 to 0.86) |
| Caseload | 0.81** (0.78 to 0.84) | 1.16 (0.91 to 1.49) | N/A | 0.13** (0.12 to 0.14) |
| Liaison | | | | |
| F2f contacts | 0.95 (0.84 to 1.07) | 0.72 (0.19 to 2.76) | 0.46** (0.39 to 0.54) | 0.60** (0.47 to 0.78) |

| | Site A | Site F | Site G | Site J |
|---|---|---|---|---|
| | Lift of the lockdown versus pre lockdown | Lift of the lockdown versus pre lockdown | Lift of the lockdown versus pre lockdown | Lift of the lockdown versus pre lockdown |
| Measures | IRR (95% CI) | IRR (95% CI) | IRR (95% CI) | IRR (95% CI) |
| Non-f2f contacts | 1.94** (1.48 to 2.54) | 17.87** (10.08 to 78.09) | 5.26** (3.42 to 8.07) | 1.34 (0.91 to 1.98) |
| F2f and non-f2f contacts | 1.03 (0.92 to 1.16) | 2.31 (0.55 to 9.71) | 0.88 (0.74 to 1.04) | 0.77** (0.62 to 0.95) |
| Cancelled appointments or DNAs | 0.69 (0.43 to 1.10) | 0.11 (0.02 to 0.70) | 1.48 (0.91 to 2.41) | 1.12 (0.61 to 2.05) |
| Caseload | 0.95** (0.94 to 0.96) | 0.82** (0.76 to 0.89) | N/A | 0.75** (0.73 to 0.76) |
| Older adults | | | | |
| F2f contacts | 0.51** (0.41 to 0.64) | 0.04** (0.00 to 0.32) | 0.27** (0.16 to 0.44) | 0.21** (0.13 to 0.33) |
| Non-f2f contacts | 2.04** (1.62 to 2.56) | 6.60 (0.69 to 63.35) | 5.37** (2.80 to 10.29) | 5.70** (3.91 to 8.29) |
| F2f and non-f2f contacts | 0.95 (0.78 to 1.16) | 0.42 (0.04 to 4.20) | 0.95 (0.58 to 1.55) | 1.78** (1.20 to 2.64) |
| Cancelled appointments or DNAs | 1.05 (0.60 to 1.82) | 0.23 (0.02 to 2.79) | 0.37** (0.19 to 0.74) | 0.97 (0.68 to 1.37) |
| Caseload | 0.76** (0.73 to 0.79) | 0.95** (0.93 to 0.97) | N/A | 0.81** (0.79 to 0.81) |

Incidence rate ratios and 95% CIs were estimated.
*p<0.05; **p<0.001.
AMH, adult MH service; CAMHS, child and adolescent MH service; DNA, Did not attend; EIP, early intervention for psychosis service; F2f, face to face; HTT, home treatment team; ; MHA, Mental Health Act; N/A, not available data for these type of services; OA, older adult service.

'protect the NHS'. Finally, a higher risk of suicide might conceivably result from rapid social, economic and health changes.[32][33] On the other hand, it is possible that the pandemic may have had positive effects on MH and the reduced use of services might reflect reduced need—for example, arising from increased family contact at home during 'lockdown', and/or reduced exposure to social pressures outside the home, and/or a sense of a more supportive community or shared adversity. Either way, the scale of mental healthcare changes has not yet been fully quantified beyond single-site findings, although recommendations made in China for tighter admission criteria and reduced hospital outpatient visits, among others,[34] may reflect similar changes in service provision to those we report.

While it was not our intention here to investigate factors underlying the observed service changes, many are likely to be unsurprising. Clearly, a reduction in f2f clinical contacts was a likely outcome of social distancing, plus rising concern about the infection risks to patients and staff of such contacts. These were balanced by an increase in non-f2f contacts, and the only reductions after 23 March 2020 in total contacts were seen in liaison and HTT services. These also saw strongest reductions in daily caseload, which for liaison services may reflect high numbers of discharges from acute hospitals to prepare for a surge in pandemic-related demand, and possibly patient reluctance to attend emergency care. While HTT caseload reductions might reflect reduced demand, the levels and consequences of unmet need during the lockdown require further evaluation. Of interest, rates of cancelled or non-attended appointments did not change significantly for most services. On the face of it, this is reassuring, given the sizeable switch towards non-f2f contacts. However, the findings should be viewed with some caution, as it is not yet clear whether unsuccessful attempts at non-f2f contacts are recorded in the same way as a non-attended f2f appointment. Furthermore, longer-term trends beyond the rather unusual context of the 'first-wave' lockdown need additional evaluation.

### Between-site variation

An important feature of the findings was their heterogeneity between sites. Mainly this reflected magnitude rather than direction of effect: for example, all sites saw increases in deaths around lockdown initiation (figure 1, online supplemental table 1), although IRRs varied in size. This heterogeneity may reflect site differences in patient demographics and pandemic timing across the country. IRRs for inpatient caseload changes at the same transition point may reflect differences in facilities—for example, strongest caseload reductions were seen at site A which had highest pre lockdown levels (table 1). On the other hand, reductions in HTT caseloads were weakest at site H where pre lockdown caseloads were highest and liaison caseload reductions differed substantially between sites (B and H) with similar pre lockdown caseloads; this might reflect local differences in acute hospital practice/demand and indicates scope for further investigation.

## Causality considerations

It is challenging to disentangle the effect of the ongoing COVID-19 pandemic itself from those of social distancing/lockdown policies (or other psychosocial consequences) on MH service activity and mortality. Our estimates retrieved from RDiT are of a compound effect: the causal effect of the 23 March 2020 lockdown announcement on MH service activity and any unobserved sorting/anticipation/adaptation/avoidance effects that may exist but cannot be tested for. The extent to which the results should be interpreted solely as the causal treatment effect of interest depend on the likelihood of other influences. For example, observed increases in mortality are likely to represent effects of the ongoing pandemic rather than lockdown policy, whereas for the service activity measures, adjustment for national mortality (sensitivity analysis 3) is less of an issue and we consider those results to be robust. However, we did not attempt to capture the variation in the timing or magnitude of infection and mortality rates across the UK.

The estimation of mortality increase in our analysis is clearly reflecting a before–after comparison with no population control and is also restricted to all-cause mortality as an outcome with no information available across sites on causes of death responsible. However, changes in all-cause mortality are as valid as those of specific COVID-19 mortality because the former encompasses a wider range of impacts beyond the direct effects of the infection, including adverse consequences of reduced access to medical care as well as suicide mortality as a consequence of reduced MH support. A report from one of the constituent sites indicates that most of the excess mortality in MH service users during March 2020 to June 2020 was from deaths attributed to COVID-19, as well as small excesses in deaths with dementia as an underlying cause.[35] In that study, there was also an excess in unexplained deaths compared with previous years, many of which were likely to involve cases awaiting inquests, although these will not necessarily result in suicide verdicts, and estimates to date (for the UK as well as a number of other nations) have not indicated increased suicide rates, at least in the early stages of the pandemic,[36] and non-fatal self-harm presentations have also been reported as relatively low during that early period.[18 37]

## Strengths and limitations

Study strengths include the use of relatively 'real-time' mental healthcare data from 10 large providers or routinely collected datasets, allowing investigation of changes in service activity following dramatic and rapid transitions at a national scale. The size of the data permitted precise estimates and the relative consistency in the direction of findings supports generalisability. Here, we report on what might be termed a 'natural experiment'; while individuals were not randomly assigned to experience lockdown or not, assignment is assumed to be quasi-random for observations close to the cut-off . Our regression discontinuity design approach

allows valid causal effects to be identified[38] and has been previously employed with electronic health record data.[39] Limitations include that data were drawn from specific services of interest and do not reflect the full activity of the 10 sites; they were also combined by service type and we did not seek to investigate within-service (team-level) variation. Daily contact numbers were quantified from structured fields in electronic health records and might reflect recording behaviour rather than activity levels (eg, if multiple contacts were subsumed within one entry); also, the dichotomy between f2f and non-f2f contact is a relatively crude one and it neither reflects the quality or depth of assessments being recorded nor the platform that was used (eg, telephone vs video). While we sought to harmonise data extractions as much as was feasible, there were inevitable differences in data availability between sites and likely variation in some measurements (eg, in the completeness of mortality data in clinical systems), although we believe there should be robust within-site consistency across the time periods examined. Some IRR heterogeneity may reflect differences in service provision (eg, acute hospital beds covered by a given liaison service). Clearly, the observation period was limited in duration and there will be a continuing need to collect data to evaluate potential longer-term consequences of both the first and subsequent pandemic waves. Finally, it is important to emphasise that findings here represent quantitative estimates of care provision, rather than care as experienced by patients; there is therefore an important need to evaluate patient satisfaction and perception of care and changes in care delivery before drawing conclusions regarding beneficial or adverse consequences.

## Implications

The alterations in MH service activity observed here have been profound and reflect the very limited time period over which to implement responses to the COVID-19 'lockdown' policy. The sharp downturn in inpatient admissions and f2f community contacts are potentially concerning, as is the apparently much longer-term shift in balance from f2f to virtual contacts which remains a marked but unevaluated change in care. On the other hand, as argued above, reduced inpatient admissions and crisis service contact might potentially reflect a reduced demand due to better MH, and virtual contacts might represent an option of care with potential advantages in the future, extending opportunities for patient choice and reducing travel and time costs for both patients and clinical services. Predicting the continuing impact of COVID-19 and related infection-control policies on MH service provision and demand remains challenging. For example, while national 'shared' emergencies may be experienced as less isolating for people with mental disorders and some consequences may be less problematic than feared,[32] treatment pathways are likely to have been disrupted significantly, particularly since social distancing/lockdown policies were imposed (continuously or intermittently) over many months. It is therefore imperative that national and international collaborations are established to monitor MH service needs and experiences of

care as closely as possible, to identify adverse outcomes early enough for rapid intervention—in particular, to understand how people with new and existing severe MH conditions can best be supported in a pandemic, given the need for socially distanced care input.

**Author affiliations**
[1]Department of Biostatistics and Health Informatics, King's College London Institute of Psychiatry, Psychology and Neuroscience, London, UK
[2]South London and Maudsley NHS Foundation Trust, London, UK
[3]Department of Psychological Medicine, King's College London Institute of Psychiatry, Psychology and Neuroscience, London, UK
[4]Southern Health NHS Foundation Trust, Southampton, UK
[5]Faculty of Medicine, University of Southampton, Southampton, UK
[6]Lancashire and South Cumbria NHS Foundation Trust, Preston, UK
[7]Department of Psychiatry, School of Clinical Medicine, University of Cambridge, Cambridge, UK
[8]Liaison Psychiatry Service, Cambridgeshire and Peterborough NHS Foundation Trust, Cambridge, UK
[9]Camden and Islington NHS Foundation Trust, London, UK
[10]Department of Psychiatry, University of Oxford Medical Sciences Division, Oxford, UK
[11]Oxford Health NHS Foundation Trust, Oxford, UK
[12]Cumbria, Northumberland, Tyne and Wear NHS Foundation Trust, Newcastle upon Tyne, UK
[13]Usher Institute of Population Health Sciences & Informatics, University of Edinburgh Division of Medical and Radiological Sciences, Edinburgh, UK
[14]Population Data Science, Swansea University Medical School, Swansea, UK
[15]Department of Psychiatry, University of Oxford, Oxford, UK
[16]Cambridgeshire and Peterborough NHS Foundation Trust, Fulbourn, UK
[17]Division of Psychiatry, University of Edinburgh Division of Medical and Radiological Sciences, Edinburgh, UK
[18]Division of Psychiatry and Applied Psychology, University of Nottingham Faculty of Medicine and Health Sciences, Nottingham, UK
[19]Nottinghamshire Healthcare NHS Foundation Trust, Nottingham, UK
[20]Division of Psychiatry, University College London Faculty of Medical Sciences, London, UK
[21]Lothian Primary Care NHS Trust, Edinburgh, UK

**Acknowledgements** This work uses data provided by patients and was collected by the NHS as part of their care and support. The authors would also like to acknowledge all data providers who made anonymised data available for research. The authors wish to acknowledge the collaborative partnership that enabled acquisition and access to the deidentified data, which led to this output and which was facilitated by MQ and the National Institute for Health Research Mental Health Translational Research Collaborative.

**Contributors** The study was initially conceived by RSt, IB and SL and further planned and designed by all authors. All authors carried out the study and approved the final version of the manuscript. Data analyses were carried out by IB, SL, AJ and SCL. The manuscript was initially drafted by IB and RSt with input on successive drafts from DB, JB, PB, MB, RC, SC, KC, AC, SD, PH, CAJ, AJ, DWJ, SCL, JL, AM, NN, DO, PP, SR, TS, RSo, RW and SL.

**Funding** Regarding relevant background infrastructure funding, RSt was part funded by: (1) the National Institute for Health Research (NIHR) Biomedical Research Centre (BRC) at the South London and Maudsley NHS Foundation Trust and King's College London; (2) a Medical Research Council (MRC) Mental Health Data Pathfinder Award to King's College London; (3) an NIHR Senior Investigator Award and (4) the NIHR Applied Research Collaboration South London (NIHR ARC South London) at King's College Hospital NHS Foundation Trust. IB and SL were supported by the NIHR BRC at South London and Maudsley NHS Foundation Trust and King's College London and by the NIHR Applied Research Collaboration South London (NIHR ARC South London) at King's College Hospital NHS Foundation Trust. RC's research was funded by the MRC (grant MC_PC_17213) and the NIHR Cambridge BRC. AC was supported by the NIHR Oxford Cognitive Health Clinical Research Facility, by an NIHR Research Professorship (grant RP-2017-08-ST2-006), by the NIHR Oxford and Thames Valley Applied Research Collaboration and by the NIHR

Oxford Health BRC (grant BRC-1215-20005). AJ was part funded by MQ ADP and an MRC Mental Health Data Pathfinder Award to Swansea University. AJ and SCL were part funded by Health and Care Research Wales National Centre for Mental Health. DO was supported by the NIHR BRC at University College London Hospitals and by the National Institute for Health Research ARC North Thames. Additional infrastructure funding was provided by the MRC Mental Health Data Pathfinder Award to University of Edinburgh (MC_PC_17209). The collaboration providing Wales data was led by the Swansea University Health Data Research UK team under the direction of the Welsh Government Technical Advisory Cell and includes the following groups and organisations: the Secure Anonymised Information Linkage Databank, Administrative Data Research Wales, NHS Wales Informatics Service, Public Health Wales, NHS Shared Services and the Welsh Ambulance Service Trust and MRC grant MR/V028367. This study was additionally supported by the NIHR Mental Health Translational Research Collaboration. The views expressed are those of the authors and not necessarily those of the UK National Health Service, the NIHR or the UK Department of Health.

**Competing interests** RC consults for Campden Instruments Ltd. and receives royalties from Cambridge University Press, Cambridge Enterprise and Routledge. DB reports royalties from Wiley publishers. AC reports research funding from Angelini Pharma and personal fees from INCiPiT, CARIPLO Foundation and Angelini Pharma. AM reports research funding from The Sackler Trust and personal fees from Janssen and Illumina. PP reports research funding from Novo Nordisk and royalties from John Wiley & Sons. SR reports education support from Janssen, Otsuka and Lundbeck. RS reports research support from Janssen, GSK and Takeda and royalties from Oxford University Press.

**Patient consent for publication** Not required.

**Ethics approval** The authors assert that all procedures contributing to this work comply with the ethical standards of the relevant national and institutional committees on human experimentation and with the Declaration of Helsinki 1975, as revised in 2008. The extractions were carried out as service evaluations (http://www.hra-decisiontools.org.uk/research/) with appropriate local information governance approval. Those for Cambridge and Peterborough and South London and Maudsley sites used CRATE and CRIS data platforms, respectively, which have been preapproved for both service evaluation and research purposes (REC references 17/EE/0442 and 18/SC/0372, respectively). Those for Wales sites received Information Governance Research Panel approval (SAIL 0911).

**Provenance and peer review** Not commissioned; externally peer reviewed.

**Data availability statement** All data relevant to the study are available from the corresponding author on request.

**ORCID iDs**
Robert Stewart http://orcid.org/0000-0002-4435-6397
Rudolf Cardinal http://orcid.org/0000-0002-8751-5167
Andrea Cipriani http://orcid.org/0000-0001-5179-8321
Caroline A Jackson http://orcid.org/0000-0002-2067-2811
Ann John http://orcid.org/0000-0002-5657-6995
Andrew McIntosh http://orcid.org/0000-0002-0198-4588

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
