## [Reviewer comments · BMJ Open]

ARTICLE DETAILS

TITLE (PROVISIONAL)	Changes in daily mental health service use and mortality at the commencement and lifting of COVID-19 'lockdown' policy in ten UK sites: A regression discontinuity in time design
AUTHORS	Bakolis, Ioannis; Stewart, Robert; Baldwin, David; Beenstock, Jane; Bibby, Paul; Broadbent, Matthew; Cardinal, Rudolf; Chen, Shanquan; Chinnasamy, Karthik; Cipriani, Andrea; Douglas, Simon; Horner, Philip; Jackson, Caroline A.; John, Ann; Joyce, Dan W.; Lee, Sze; Lewis, Jonathan; McIntosh, Andrew; Nixon, Neil; Osborn, David; Phiri, Peter; Rathod, Shanaya; Smith, Tanya; Sokal, Rachel; Waller, Rob; Landau, Sabine

VERSION 1 – REVIEW

REVIEWER	Chakraborty, Nandini Leicestershire Partnership NHS Trust
REVIEW RETURNED	27-Feb-2021

GENERAL COMMENTS	1. 'To investigate the impact of the COVID-19 pandemic on mental health services'. is not a clear objective. This could be interpreted in various ways- impact on patient outcomes (suicides, admissions, crises, satisfaction/complaints, waiting times), service use (more referrals) or resources (staff being deployed, reduced f2f contact, longer time for routine work). The authors have focussed on service use and added mortality statistics. The title of the study captures this more accurately than the statement of the objectives and hence the objectives should be changed to reflect consistency in what this study aimed to look at. 2. The mortality statistics do not convey a significant meaning in this study. There are no causes of deaths mentioned. Hence one cannot make any deductions about whether suicides increased at commencement of lockdown. It appears likely that the authors are talking about COVID related deaths. Unless this is compared against mortality statistics in general or other hospital settings, on its own, there is not clear message from this finding. 3. I have attached a set of references which I think are current and relevant, in addition to what has already been mentioned. I think these are important for incorporating in the discussion. 4. The results do not discuss the DNA rates, though these are stated in the tables. Whether DNA rates changed with non f2f appointments is an important point of discussion. 5. The discussion starts with concerns around reduction in hospital admission rates and f2f contacts. I think there should be a more two-way consideration here. Could there have been any positive factors associated with reduced hospital admission? The fact that people were not pressured to go out if they already found it stressful due to anxiety and socialising problems? The fact that families were more together and might have in some instances led
--

	to more support, supervision of medication for example? Post lockdown admission rates increased but not superseding pre-COVID numbers. Similarly for non f2f contacts, which was only to be practically expected-does this offer us added options for patient choice in future? I think that is an important point for consideration as we look at the future of post-lockdown/post-COVID services. In fact the authors note that total number of contacts increased. Do non f2f contacts provide an important resource for future, cutting down travel and time costs for both patients and professionals? 6.What we are missing in literature is a sizeable study of patient satisfaction and perception of care especially non f2f contacts in mental health services. Some consideration of this point in discussion is worth.
--	---

REVIEWER	Brodeur, Abel University of Ottawa
REVIEW RETURNED	24-Mar-2021

GENERAL COMMENTS	The authors rely on a method which is very popular in economics and political science, but not in the field of health sciences. They are far from the frontier of research from a methodological point of view in economics, but clearly at the frontier for health sciences. This makes it difficult to assess the contribution of this paper. I would like the authors to conduct additional empirical exercises before making a final decision on this manuscript. In my comments to the authors, I point out some issues that need to be taken into account. This is an incomplete list of suggestions that I have, but would prefer to see an updated version of this manuscript prior to making more technical comments. These comments are in line with important developments made for RDD since the early 2010s. Here are few examples: the inference is wrong, the authors do not plot the point estimates with the full set of controls over time, they do not discuss the functional form, etc.
---

VERSION 1 – AUTHOR RESPONSE

Reviewer: 1

1.'To investigate the impact of the COVID-19 pandemic on mental health services'. is not a clear objective. This could be interpreted in various ways- impact on patient outcomes (suicides, admissions, crises, satisfaction/complaints, waiting times), service use (more referrals) or resources (staff being deployed, reduced f2f contact, longer time for routine work). The authors have focussed on service use and added mortality statistics. The title of the study captures this more accurately than the statement of the objectives and hence the objectives should be changed to reflect consistency in what this study aimed to look at.

We thank the reviewer for their comments. We have altered the sentence in the Objectives section of the Abstract accordingly.

2. The mortality statistics do not convey a significant meaning in this study. There are no causes of deaths mentioned. Hence one cannot make any deductions about whether suicides increased at commencement of lockdown. It appears likely that the authors are talking about COVID related deaths. Unless this is compared against mortality statistics in general or other hospital settings, on its own, there is not clear message from this finding.

We accept the limitations of what is essentially a before-after analysis of mortality for each site controlled for the previous year, although we do believe it has some meaning. We do not have access to cause of death data across all sites, and were therefore not able to analyse this, although findings have been reported on excess deaths by cause for one of the participating sites. We have added text to the Discussion (6th paragraph) citing and summarising this single-site information (representing, as far as we are aware, the only UK data so far on the matter), the importance of all-cause mortality as an outcome as well as COVID-specific mortality, and recent opinion on the lack of evidence for raised suicide rates so far. We should stress that we weren't seeking to address the question about suicide rates around lockdown (as we knew that this would not be possible) and we have double-checked the manuscript to ensure that this wasn't inadvertently implied.

3. I have attached a set of references which I think are current and relevant, in addition to what has already been mentioned. I think these are important for incorporating in the discussion.

We are grateful for these suggestions and have added citations to all of these, and additional accompanying text, in relevant sections of the manuscript.

4. The results do not discuss the DNA rates, though these are stated in the tables. Whether DNA rates changed with non f2f appointments is an important point of discussion.

We accept this suggestion, and have added text in the Discussion (3rd paragraph) considering in more detail the general lack of change in DNA rates. As mentioned, we would recommend caution in interpretation, as it is not clear whether unsuccessful non-f2f contacts are recorded in the same way as non-attended f2f appointments.

5. The discussion starts with concerns around reduction in hospital admission rates and f2f contacts. I think there should be a more two-way consideration here. Could there have been any positive factors associated with reduced hospital admission? The fact that people were not pressured to go out if they already found it stressful due to anxiety and socialising problems? The fact that families were more together and might have in some instances led to more support, supervision of medication for example? Post lockdown admission rates increased but not superseding pre-COVID numbers. Similarly, for non f2f contacts, which was only to be practically expected-does this offer us added options for patient choice in future? I think that is an important point for consideration as we look at the future of post-lockdown/post-COVID services. In fact the authors note that total number of contacts increased. Do non f2f contacts provide an important resource for future, cutting down travel and time costs for both patients and professionals?

We agree with these matters raised. We have added text to the Introduction, raising the issue of potential positive effects, and have revisited this in the Discussion section towards the end of the second paragraph, and in the final Implications paragraph.

6. What we are missing in literature is a sizeable study of patient satisfaction and perception of care especially non f2f contacts in mental health services. Some consideration of this point in discussion is worth.

Again, we agree with this and have added the need for more research on this question to the end of the strengths and limitations paragraph.

Reviewer: 2

This paper investigates the impact of the pandemic and lockdowns on deaths, referrals and discharges, mental health, and other outcomes for ten UK providers.

My main issues with the paper are the following:

1. The empirical analysis tackles the causal impact of the national lockdown, but the stated objective is to "investigate the impact of the COVID-19 pandemic on mental health services". Disentangling the impact of the pandemic and the lockdown is very important, and clearly this is what the authors are trying to do as they control for COVID-19 deaths in some models as a sensitivity analysis. I would rewrite the stated objective (and the rest of the paper) as follows "investigate the impact of the national lockdown on mental health services".

We thank the reviewer for their helpful comment. We have altered our stated objectives in the Abstract and have altered text within the rest of the manuscript which we now hope describes better the objectives of the study. We have not used precisely the suggested wording because of the stipulation of the other reviewer to replace the "investigate the impact" terminology with something more precise. We hope that the result is satisfactory to both reviewers.

2. Given the fact that the authors are trying to identify the causal impact of the lockdown, they need to clearly think about the counterfactual. I think the counterfactual they have in mind is how the outcome variables would have changed over time if the lockdown would not have been implemented. In order to evaluate the impact of the lockdown, they rely on a RDD-DID framework. This method identifies a local average treatment effect (LATE) of the lockdown under some assumptions. Here are some comments related to the method:

-One of the main advantages of RDD over other non-experimental methods is that the results can be shown with graphs. I thus want the authors to show in supplementary figures how the dependent variables vary over time pre/post lockdown for all sites combined. This exercise needs to be done for all the main dependent variables rather than just Deaths. Moreover, this figure should plot the point estimates rather than the raw data, with the full set of control variables. This is standard for RDDs.

We thank the reviewer for their suggestion. We decided to employ for our analysis a two-stage regression discontinuity in time design. First, we explored the causal effect of COVID-19 'lockdown' policy on mortality and mental health service activity with the use of negative binomial regression models within each site separately. Second, we provided an overall pooled effect estimate with the use of random-effect meta-analysis.

Thus, we would prefer to present the point estimates for each site separately as we did in supplementary figure 5 in our original submission. Due to the large number of graphs we decided to select 4 sites and 1 outcome.

Nevertheless, we have added the additional graphs requested by the reviewer for all sites combined at the end of this response (Figure A). We would prefer not to include them in the supplementary material as we feel that this might confuse the reader. The reason is that when we combined all the data across the sites, we re-ran negative models for all sites combined (and not separately), which is a different procedure and model than the two-stage one that we described above.

-The inclusion of more controls for the severity of the pandemic (lagged daily cases, media coverage, other non-pharmaceutical policies, etc.) is crucial for identification. For instance, are the authors capturing the impact of the lockdown or simply the impact of national media coverage of the pandemic? For media coverage, they could simply rely on a major newspaper or tv network and control for the share of news stories on covid. This is quite standard and straightforward for the US (e.g., <https://tvnews.vanderbilt.edu/>), but I am unfortunately not familiar with media data in the UK.

We would like to thank the reviewer for their suggestion. Unfortunately, we could not control for the share of news stories on covid as there are not available raw media data in the UK to the best of our knowledge. As a proxy for the share of news on covid we decided to further adjust our models for Google Trends in the UK for covid-19 related searches as part of our sensitivity analysis (please see the 5th sensitivity analysis listed in the Statistical Analysis section) and we also added an additional table in our supplementary material (Supplementary Table 7) with the results of this analysis (also summarised in the final paragraph of the Results).

-The following controls should also be included: year, week and day of the week fixed effects as well as lagged number of new deaths (or cases) from Covid-19 per capita for each region/cluster (i.e., cities/communities near the site).

We thank the reviewer for their suggestions. We originally made an attempt to address temporal trends. We added year and weekday as described in the technical appendix of the supplemental material. We decided not to include week as a categorical variable due to issues of over-

parametrisation in our models which also lead to unreliable estimates and 95% confidence intervals from our models.

We also made an attempt to adjust for the severity of the pandemic by adjusting for lagged number of daily and weekly new deaths. However, we were not able to do this per capita for each region as the catchment areas of the mental health services do not adequately map on to regions for which these data are available.

-Inference: The authors should cluster their standard errors (CI) at the site-level or at the very least at the day level. Given the small number of clusters (i.e., sites), I would also suggest bootstrapping the standard errors (CI).

We would like to thank the reviewer for their suggestion; however, we think the clustering issue does not affect our analysis approach for reasons outlined below.

As mentioned, we decided to employ for our analysis a two-stage regression discontinuity in time design. First, we used negative binomial regression models within each site separately to explore the causal effect of the pandemic on mortality and mental health service activity. Second, we provided an overall pooled effect estimate with the use of random-effect meta-analysis.

Thus, there is no clustering that we need to take into account during the first stage, as we are not analysing aggregated data from all sites, but for each site separately.

We considered that our two-stage approach was more appropriate in this context than an analysis method which attempts to model the time series data from all sites simultaneously (e.g. mixed models, clustering of standard errors) due to: i) complexity of time series data within each site which is difficult to model with combined data; ii) better visual representation of results; iii) estimation of heterogeneity of the causal effect across the different sites (e.g. forest plots).

3- Can the authors provide more information on their dependent variable 'deaths'. Do you have information on the causes of death, and could you exploit those? Showing that non-covid deaths are increasing would be important here.

We thank the reviewer for their suggestion. As mentioned in our response to the other reviewer, unfortunately, we do not have data on causes of deaths for a sufficient number of sites to permit analysis, and this is therefore highlighted as a topic for future research. As described above, we have added text to the Discussion section (paragraph 6) considering mortality in more detail. In now-cited output from one of our constituent sites (the only one, we believe, with cause of death data), most of the excess mortality compared to previous years was attributable to Covid, although there was also excess in mortality with dementia as an underlying cause of death, and mortality from as-yet-undetermined causes. Although the latter will include suicide deaths awaiting inquest verdicts, it will

also include unexplained natural causes, and we cite recent opinion on national data that suicide mortality has not yet shown an increase.

Figure A. Number of deaths and mental health service activity per day before and after lockdown announcement for sites A, B, C, D, E, F, G, H and J combined during the period of 1st January 2020 to 31st May 2020. The vertical axis shows the daily number of deaths and mental health service activity in the days before (orange dots) and after (red dots) the lockdown order was announced on 23rd March and after the lift (purple dots) of the lockdown order announced on 10th May. The green dots represent the number of daily deaths and mental health service activity during the period of 1st January 2019 to 31st May 2019. The lines represent the predicted values of our fitted model.

VERSION 2 – REVIEW

REVIEWER	Chakraborty, Nandini Leicestershire Partnership NHS Trust
REVIEW RETURNED	12-May-2021
GENERAL COMMENTS	This was a well written, clearly described article and I have no further changes to propose. The limitations are already listed by the authors. The findings are observational and as stated causality is difficult to assign clearly. It still remains clinically relevant and forms a base for future studies and service evaluations which should continue.